# Development and Optimization of a Germination Assay and Long-Term Storage for *Cannabis sativa* Pollen

**DOI:** 10.3390/plants9050665

**Published:** 2020-05-23

**Authors:** Daniel Gaudet, Narendra Singh Yadav, Aleksei Sorokin, Andriy Bilichak, Igor Kovalchuk

**Affiliations:** Department of Biological Sciences, University of Lethbridge, Lethbridge, AB T1K 3M4, Canada; daniel.gaudet@uleth.ca (D.G.); narendra.yadav@uleth.ca (N.S.Y.); aleksei.sorokin@uleth.ca (A.S.); a.bilichak@gmail.com (A.B.)

**Keywords:** *Cannabis sativa*, pollen germination assay, cryopreservation, long-term storage, DAPI staining of germinated pollen, vegetative nuclei, sperm nuclei

## Abstract

Pollen viability and storage is of great interest to cannabis breeders and researchers to maintain desirable germplasm for future use in breeding or for biotechnological and gene editing applications. Here, we report a simple and efficient cryopreservation method for long-term storage of *Cannabis sativa* pollen. Additionally, the bicellular nature of cannabis pollen was identified using DAPI (4′,6-diamidino-2-phenylindole) staining. A pollen germination assay was developed to assess cannabis pollen viability and used to demonstrate that pollen collected from different principal growth stages exhibited differential longevity. Finally, a simple and efficient method that employs pollen combined with baked whole wheat flour and subsequent desiccation under vacuum was developed for the long-term cryopreservation of *C. sativa* pollen. Using this method, pollen viability was maintained in liquid nitrogen after four months, suggesting long-term preservation of cannabis pollen.

## 1. Introduction

Cannabis or hemp (*Cannabis sativa* L.) is an annual, primarily dioecious flowering plant. The center of origin is in Central Asia, and it has been bred for thousands of years for a variety of traits, including fiber, oil, seed and drug use [1]. Cannabis is a diploid plant (2n = 20) and males are characterized by heterogametic chromosomes (XY) with homogametic chromosomes (XX) conferring the female phenotype. Male plants produce flowers containing stamens producing pollen whereas female plants develop ovaries that produce seed following pollination. Female inflorescences are characterized by secretory hairs known as glandular trichomes, which produce a resinous mix of cannabinoids and aromatic compounds that are valued for both medical therapeutics and recreational effects [2].

Pollen viability is of great interest to breeders and researchers alike. Breeding projects may wish to store pollen for extended periods of time, where high value genetic material may be stored for future use or for biotechnological and gene editing applications that requires a quick and effective method for determining pollen viability [3,4,5]. Fluorescent stains such as fluorescein diacetate (FDA) or fluorochromatic reaction test (FCR) have been previously reported for assessing pollen viability in cannabis [3,4]. Viability is not always correlated with germination, as pollen may retain the ability to metabolize while losing its ability to germinate [5]. To better assess germination, we established a pollen germination assay (PGA) to estimate germination rates. We also adapted a DAPI (4′,6-diamidino-2-phenylindole) stain to visualize pollen pre- and post-germination, and to establish whether *Cannabis sativa* pollen was a bicellular or tricellular, which to our knowledge has not been reported in the literature. In approximately 30% of angiosperms, pollen is tricellular, with the male gametophyte sexually mature at the time of anthesis [6]. We also used the PGA to test how storage and timing of pollen collection influences germination rates. Pollen germination rates were assessed over a period when stored at 4 °C from males at different stages of floral development. Finally, we developed a simple procedure for the long-term storage of cannabis pollen using desiccation with baked whole wheat flower followed by cryopreservation, which potentially maintains long-term viability of pollen for future use.

## 2. Results and Discussion

### 2.1. Optimization of Pollen Germination for PGA

To obtain a representation of the germination profile, a time-lapse of a pollen germination assay (PGA) was evaluated using microscopy. We observed the germination profile for 6 h with 30 min interval. The final germination was calculated after 16 h incubation. Germination started within 30 min with extending pollen tubes clearly visible (Figure 1 and Appendix A).

Cannabis pollen readily germinated in the Pollen Germination Media (PGM). PGM was evaluated both as a liquid and a solid media (1% agar). Germination rates were comparable in both media; however, pollen tubes were not as easily imaged under the microscope when germinated on solid agar medium (data not shown). For this reason, we opted for performing the PGA using liquid media. Of the different concentrations of pollen tested, 0.1 mg/mL provided the clearest imaging of germination, as higher concentrations resulted in crowding in the test well that reduced visibility (Figure 2). Additionally, in the highest density treatment, germination was adversely affected and made it difficult to accurately quantify germination percentage (Figure 2).

### 2.2. Pollen Collected at Different Principal Growth Stages Exhibits Different Longevity

To establish how cannabis pollen germination rates change over time, we tested the pollen in a pollen germination assay after storage at 4 °C. Because pollen collected from different principal growth stages may affect germination rates, we collected pollen from male flowers at four different points during floral development to cover the entirety of anthesis (Appendix A).

We compared the loss of viability of cannabis pollen collected from the four different points during flower development over the course of 21 days. The rate of germination at T0 was 33% for Early (62), 46% for Mid (64), 50% for Mid-Late (65) and 41% for Late (64) stage (Figure 3). All stages lost viability after only one week at 4 °C storage, except Mid (64) (Figure 3). After 21 days storage at 4 °C, pollen collected from Early (62), Mid-Late (65) and Late (67) stages, lost their viability (approached 0% germination). However, pollen collected from the Mid flowering stage (64) retained viability the longest with 22% of pollen grains successfully germinated after 21 days storage at 4 °C (Figure 3). This suggested that an optimal growth stage for pollen collection is around the developmental stage (64), whereas the loss of pollen viability may begin while the pollen is still present in the anthers. Pollen collected earlier, at developmental stage 62 may not have fully matured, resulting in a lower germination percentage (Figure 3).

### 2.3. DAPI Staining Revealed Bicellular Nature of Cannabis Pollen

While the fluorescein diacetate (FDA) stain is routinely used for viability tests, it is not ideal for visualizing the nuclei in pollen cells. In order to establish whether cannabis pollen was bicellular or tricellular, we performed a DAPI stain on germinating cannabis pollen. Prior to pollen tube germination, the brighter, more compact sperm nucleus and the diffuse vegetative nucleus were visible (Figure 4A,B). The brighter staining in the sperm nucleus represents the more condensed state of chromatin compared to the more transcriptionally active vegetative nucleus. Following pollen tube germination, both sperm nuclei are clearly visible as they descend the pollen tube (Figure 4C). This suggests that cannabis releases sexually immature pollen grains, with the second mitosis event occurring after pollen tube germination.

### 2.4. Development of a Cryopreservation Method for Cannabis Pollen

Pollen cryopreservation has been employed in a variety of agriculturally and medicinally important plant species for the preservation of elite germplasm. Numerous studies have reported the data on pollen viability under various storage conditions [5,7]. While the interaction between pollen water content and viability is complex, it is understood that optimum water content is necessary for longevity [5]. Generally, longevity is increased by lowering the temperature and moisture content. Some reports indicate a moisture optima of 15%, while higher water concentrations (above 30%) may result in rapid deterioration during cryopreservation [8]. Liquid Nitrogen (LN; −196 °C) is routinely used for cryogenic storage, as it is relatively cheap, safe and maintains a temperature where enzymatic and chemical reactions do not cause biological deterioration [9]. Cannabis pollen stored in LN without prior desiccation failed to germinate (Appendix A). Pollen cells with high moisture levels do not survive cryogenic storage, presumably due to intracellular ice formation [5]. Therefore, pollen cells need to be dried within a range where no freezable water exists without succumbing to desiccation injury. For pollen desiccation, we tested a vacuum desiccation at pressures of 5, 15 or 25 kPa for either 20 or 40 min. When pollen was desiccated prior to storage in LN, it failed to germinate (Appendix A), suggesting that desiccation alone may not be sufficient for pollen viability during cryopreservation. Therefore, in addition to desiccation, we also tested cellular cryoprotectants, such as DMSO and glycerol that have been reported to improve cell survival after cryogenic storage [10]. Desiccated cannabis pollen combined with a 10%, 20%, 30% or 60% DMSO or glycerol solution prior to being stored in LN for 24 h exhibited 0% germination (Appendix A).

Baked wheat flour has been previously suggested as a possible cryoprotectant for long term pollen storage [11]. To test whether baked wheat flour can be used as a cryoprotectant for cannabis pollen, cannabis pollen was desiccated and combined with baked wheat flour. Vacuum desiccation at a lower pressure of 5 kPa for the longest interval for 40 min, resulted in the highest germination rate after storage in LN after 24 h (Figure 5). Pollen germination did not occur at higher pressures, as the cells may have been compromised during the drying process. This treatment was used for subsequent preservation experiments where the GLM test results indicated no significant differences in germination rate between 24 h LN stored pollen and the non-LN control pollen (*p > 0.05*) that was subjected to the same desiccation protocol and combined with whole wheat flour (Figure 6). Desiccation itself caused approximately 50% reduction in germination as compared to untreated freshly harvested pollen (Figure 3 and Figure 6). Desiccated cannabis pollen combined with baked wheat flour was kept in LN for four months to test long term storage. The GLM test results indicated that there was no significant difference observed as compared to non-LN control and 24 h LN stored pollen (*p > 0.05*) (Figure 6), suggesting long term storage is a possibility under appropriate conditions. To confirm *in planta* viability of the treated cannabis pollen, the pollen/wheat flour mix was removed from LN and applied to flowering female cannabis plants. The pollination resulted in successful seed formation in all the flowers receiving treated pollen. Once the female had finished flowering, the flower material was collected and processed for seeds. Seed number, size and morphology from the cryopreserved pollen were similar to those obtained using untreated fresh pollen (Appendix A). Collected seeds were germinated to ensure viability, with no abnormalities noted.

There are several reports on pollen long-term cryopreservation including one-year viability (*Allium* sp., [12]; *Juglans nigra*, [13]; *Diospyros khaki* [14]), two years viability (*Jojoba*, [15]; *hop*, [16]), and five or more years survival (*Vitis vinifera L.*, [17]; tomato and eggplant, [18]; Maize, [19]; Gladiolus [20]). Our cryopreservation method resulted in a slight decrease in germination (but not significant, GLM test *p > 0.05*) after 24 h and four-month of LN storage (Figure 6). Hamzah and Chan (1986) suggested viability declines over a relatively short time [21]. *Hevea* pollen exhibited a decline from 20% in vitro germination after one month to 2% after five months of storage in LN. Some pine and spruce pollen stored in LN also showed a decline in viability over a 24-month period [22]. Cryopreservation of maize, lily [23] and wheat pollen [24] also exhibited a decline in viability during cryopreservation. Overall, these results suggest that periodic viability testing of cryopreserved pollen is required to ensure the future viability of stored pollen in breeding.

In conclusion, we have standardized a simple assay for quickly assessing pollen germination in *Cannabis sativa.* Through the use of DAPI staining on germinating pollen cells, we were able to track the migration of sperm nuclei descending the pollen tube. This indicates that *Cannabis sativa* releases pollen in a bicellular state, where the second mitosis event occurs after pollen tube germination. By using our PGA, we have demonstrated the loss of pollen viability over time when stored at 4 °C and suggested an optimal time during flower development for pollen collection to maximize longevity during storage. Finally, we have provided an easy protocol for cryopreservation using desiccation combined with baked wheat flour and subsequent long-term storage of cannabis pollen in liquid nitrogen.

## 3. Materials and Methods

### 3.1. Plant Material and Growth Conditions

*Cannabis sativa* plants (strain name “Spice”, THC dominant) were grown under full spectrum 300-Watt LED grow lights (PrimeGarden) with 16 h light for vegetative growth and 12 h light for flowering at 22 °C.

### 3.2. Pollen Germination Assay (PGA)

#### 3.2.1. Pollen Germination Media (PGM)

The composition of pollen germination medium was adapted from Schreiber and Dresselhau (2003) [25] with some modification. The original pollen germination medium from Schreiber and Dresselhau (2003) [25] employed 1% noble agar. In our study, we tested pollen germination media as liquid or combined with 1% agar. We found that liquid media resulted in better image acquisition and quantification of germination than solid media. Therefore, we employed liquid medium for all pollen germination experiments. During optimization of the Pollen Germination Assay (PGA), pollen concentrations of 0.1, 1 and 10 mg/mL were employed with the pollen diluted in PGM and incubated for 16 hr.

A 2× PGM contained the following: 10% sucrose (BIOSHOP), 0.005% H_3_BO_3_ (Sigma), 10 mM CaCl_2_ (BIOSHOP), 0.05 mM KH_2_PO_4_ (Merk) and 6% PEG 4000 (Fluka). After components were added to distilled H_2_O, heated on a stir plate for 10 min at 70 °C then filter sterilized. A 1× working solution was prepared fresh each day by diluting in distilled H_2_O.

#### 3.2.2. Pollen Collection and Optimization of the PGA

Pollen was obtained from flowering male *Cannabis sativa* plants using a vacuum manifold method [26]. For the standardized PGA, 10 mg of cannabis pollen was combined with 1 mL of freshly prepared 1× PGM and diluted to 0.1 mg/mL. 200 µL was then pipetted into a 24-well tissue culture plate (Flat Bottom Cell+, Sarstedt) and sealed with parafilm. Plates were incubated in the dark at 22 °C for 16 h and examined using an inverted light microscope (Zeiss Axio Observer Z1, Oberkochen, Germany). To obtain a representation of germination profile, a time-lapse of a pollen germination assay (PGA) was performed. We have observed the germination profile for 6 h with 30 min interval. The final germination was calculated after 16 h incubation.

### 3.3. Imaging and Germination Assessment

Images were taken using phase contrast at 100× magnification. For each treatment, the germination experiment was repeated at least three times with three biological replicates. For each technical replicate, 8 images were taken to get an accurate representation of germination. Pollen germination percentages were calculated by dividing the number of germinating pollen grains by the total number of pollen grains. Germination percentages for each replicate represent the averages of the eight images.

### 3.4. DAPI Staining of Cannabis Pollen to Decipher Its Bicellular or Tricellular Nature

Collected pollen was stained with DAPI (4′,6-diamidino-2-phenylindole) and imaged using an inverted fluorescent microscope (Zeiss Axio Observer Z1). The DAPI staining protocol was adapted from Backues et al. 2010 [27]. Germinated pollen was suspended in pollen isolation buffer (PIB) containing 100 mM NaPO_4_ (pH 7.5), 1 mM EDTA, 0.1% (*v*/*v*) Triton X-100 and 1 µg/mL DAPI. A drop of solution was placed on a coverslip, incubated at room temperature for 5 min and viewed with the DAPI filter set. For DAPI staining of germinated pollen, pollen germination was performed as previously described, and staining was conducted with 1 µg/mL DAPI after 16 h in PGM.

### 3.5. Pollen Collection from Different Development Stages to Assess the Loss of Pollen Viability over Time

Pollen was collected from male *Cannabis sativa* plants at different stages of the flower development. The four stages of flowering were chosen according to the BBCH (Biologische Bundesantalt, Bundessortenamt and Chemische) scale adapted for cannabis [28] and are listed as follows with the BBCH notation in brackets: Early (62), Mid (64), Mid-Late (65) and Late (67).

Following collection, an aliquot from each developmental stage was taken and used in the PGA for germination rate at time of collection (T0). The rest of the aliquots were stored in a 1.5 mL centrifuge tube at 4 °C in the dark. After one week, an aliquot from the different developmental stages was used for the PGA (T1), again after 2 weeks (T2) and again after 3 weeks (T3).

### 3.6. Cryopreservation of Pollen

Cannabis pollen submerged in Liquid Nitrogen (LN) without the use of any cryoprotectant or treatment will fail to germinate after the formation of ice crystals [29,30]. Cannabis pollen was combined with cryoprotectants DMSO or glycerol diluted to concentrations of 10%, 20%, 30% and 60% and submerged in LN. Following 24 h in LN, pollen was removed and used in the PGA as previously described.

### 3.7. Desiccation of Pollen Prior to Cryopreservation

Cannabis pollen was combined with all purpose baked wheat flour (1:10 *w*/*w*) in a 1.5 mL centrifuge tube and desiccated at 5, 15 and 25 kPa for 20 or 40 min. Following desiccation, the tube was placed in LN for four months, removed and placed at 22 °C for 10 min. The pollen/wheat flour mix was then used for the PGA, as previously described.

### 3.8. Statistical Analyses

Mean pollen germination over time at four different development stages and mean pollen germination efficiency between pollen stored in liquid nitrogen (LN) for 24 h, 4 months and non- liquid nitrogen control were calculated in an excel sheet. Data were shown as mean ±SE. The germination efficiency data were analyzed by generalized linear models (GLM) with binomial distribution (link logit) using the GLM function in software R Studio 1.2.1335.

## Figures and Tables

**Figure 1 plants-09-00665-f001:**
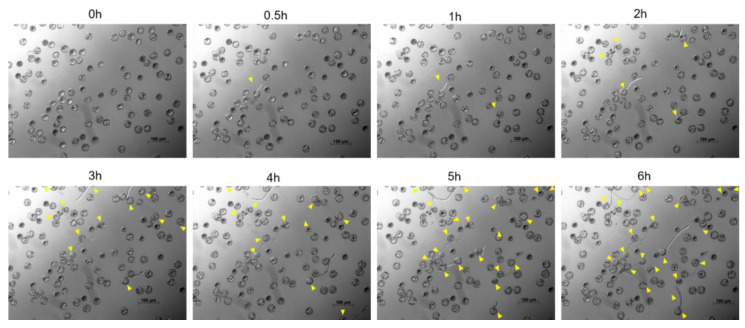
Representative photographs of cannabis pollen germination profile. Images were taken at 30-min intervals for 6 h with germinating pollen grains indicated by the yellow arrows. Germination started within 30 min with extending pollen tubes clearly visible. Images were acquired using an inverted fluorescent microscope (Zeiss Axio Observer Z1, Germany).

**Figure 2 plants-09-00665-f002:**
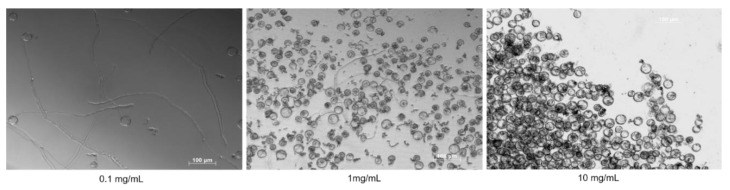
Optimization of the Pollen Germination Assay. Cannabis pollen germination in PGM at concentrations of 0.1, 1 and 10 mg/mL. Images were acquired after 16 h using an inverted fluorescent microscope (Zeiss Axio Observer Z1, Germany).

**Figure 3 plants-09-00665-f003:**
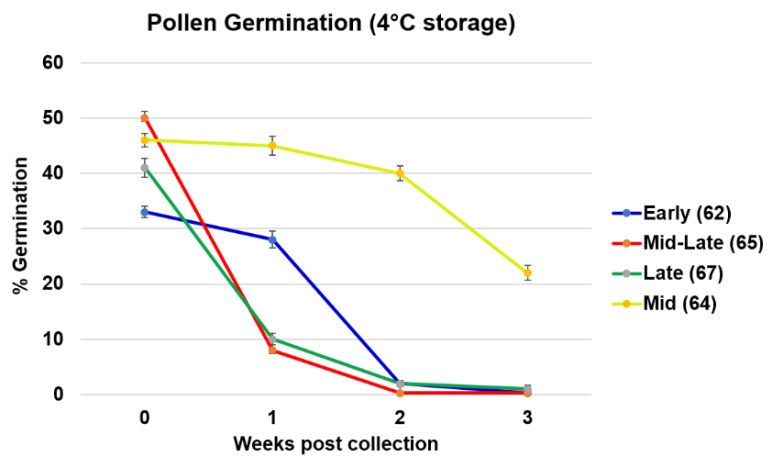
Loss of pollen viability over time. Pollen was harvested from plants at four different developmental stages then stored at 4 °C for one to three weeks. Viability was determined via pollen germination assay. Data were shown as mean ± SE (*n = 9*).

**Figure 4 plants-09-00665-f004:**
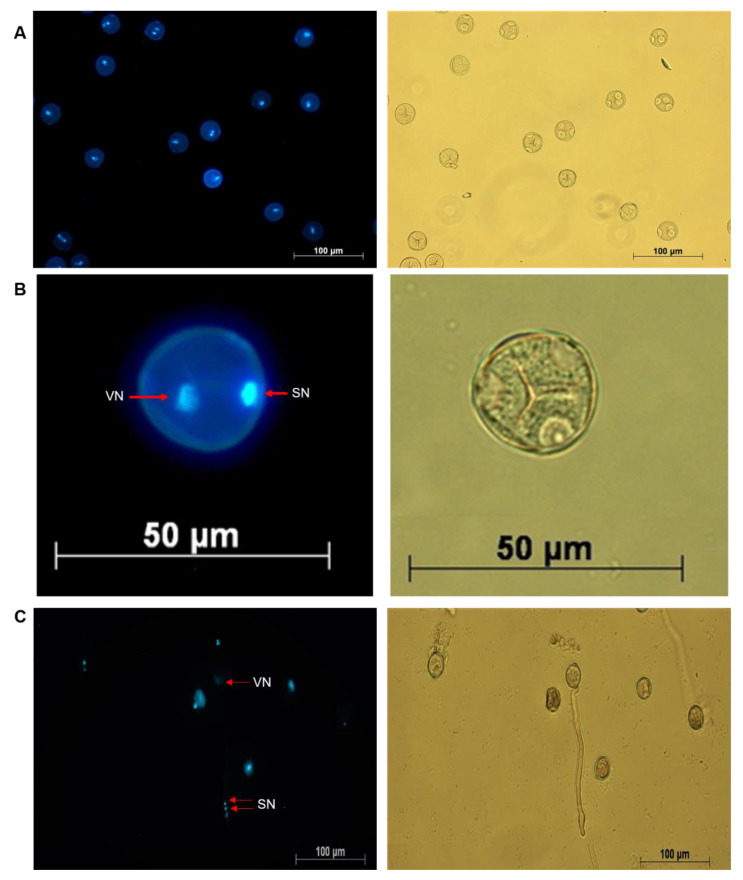
Visualization of cannabis pollen at different stages of germination using DAPI staining. Both the sperm (SN) and the vegetative nuclei (VN) are visible at the bicellular stage prior to pollen tube germination (**A**, **B**). Image (**C**) represents a germinated cannabis pollen cell.

**Figure 5 plants-09-00665-f005:**
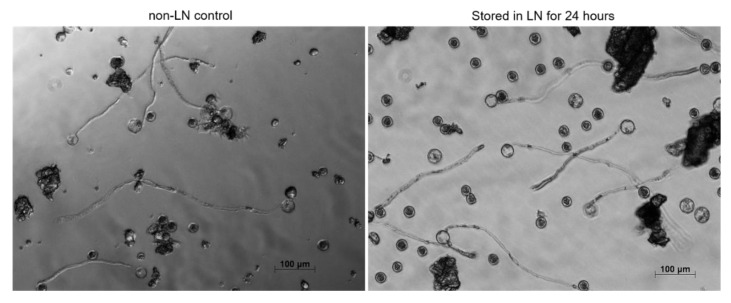
Representative photographs from pollen germination assay (PGA) of pollen stored for 24 h in liquid nitrogen (LN). Desiccated cannabis pollen mixed with 1:10 wheat flour and stored in liquid nitrogen (LN). Non-LN control (control was subjected to the same desiccation combined with whole baked wheat flour). Pollen flour mix was diluted to 0.1 mg/mL in PGM and used for PGA.

**Figure 6 plants-09-00665-f006:**
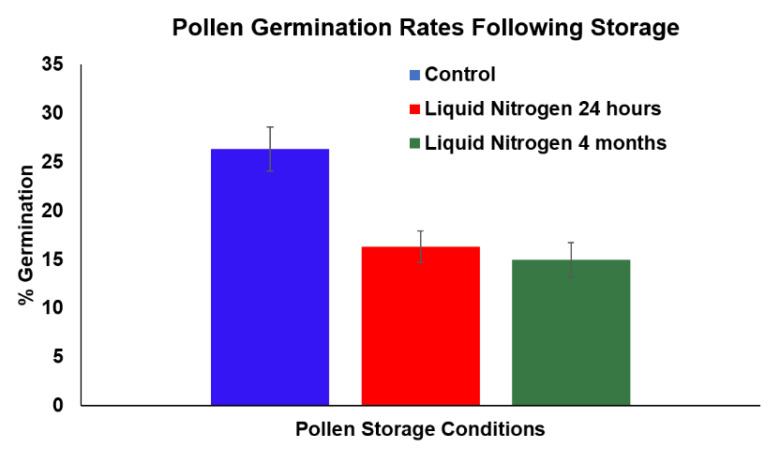
Comparison of pollen germination efficiency between desiccated pollen combined with whole baked wheat flour and stored in liquid nitrogen (LN) for 24 h or 4 months and non- liquid nitrogen control (control was subjected to same desiccation protocol combined with whole baked wheat flour). Data were shown as mean ± SE. The germination efficiency data were statistically analyzed by generalized linear models (GLM) with binomial distribution (link logit) using the GLM function in software R Studio 1.2.1335.

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
