# Peer review of "Development and Optimization of a Germination Assay and Long-Term Storage for Cannabis sativa Pollen"

_plants, 2020, doi:10.3390/plants9050665_

Round 1

Reviewer 1 Report

Dear authors I have appreciated your article that could be useful for researchers and breeders. I suggest you few corrections that you can find in the text Revised.

Reviewer 2 Report

Hi authors,

Your manuscript on pollen germination and storage was generally well-written. However, two keys concerns are:

  1. No mention of statistical analyses in Materials and Methods
  2. Frequent use of "data not shown" in Results and Discussion
    - If readers combine line 177 and line 181, this manuscript essentially shows no data on failed germination whether pollen stored in LN without prior desiccation (line 177) or with (line 181)! It is unscientific to report such statements when neither results were shown.
    - Lines 184-186: Similarly, data from combination with DMSO or glycerol solution were not shown, conveying no empirically verifiable results to readers.

Other minor tweaks to incorporate:

  • Unabbreviate DAPI at first mention (Abstract and Main Text)
  • Lines 14-20: Change to passive voice
  • Use proper centigrade symbol throughout manuscript
  • Use proper multiplication symbol (instead of letter X) to denote multiplication factor for solvent concentration
  • Line 58: Substantiate this sentence
  • Every time a literature-reported method is used and cited, provide at least a brief summary of what was implemented
  • Lines 110 to 112: In Materials and Methods section instead
  • Figures 3 and 6: Standardize y-axis (% Germination or Percent Germination)
  • Figure 6: Unabbreviate LN; note that all figures should be able to standalone
  • Figures 3 and 6: Error bars must be analyzed and reported as either "Standard Error" or ideally "95% Confidence Interval"
  • Line 224: " ... required to ensure future viability of stored ..."

Reviewer 3 Report

MS ID; plants-794967

In this MS, authors proposed a protocols and result data for pollen germination and long-term storage conditions of Cannabis sativa pollen. I think that this type of research is valuable not only to breeders of Cannabis plants, but also to researchers involved in plant reproductive research. Research were well conducted. I would like to recommend publication of the paper after the authors have addressed the concerns which I have stated below. And I wrote it in the miner comment, but there are many mistakes especially in the Ref part. Authors proceed to check the entire MS.

Major 1.

If possible, add some representative photos of the plant material used in this MS. In particular, the developmental stages have a severe effect on pollen viability (Fig. 3), so I think this pictures will bring important information to the readers.

Major 2.

Line 200, “To confirm in planta viability of the treated cannabis pollen, the pollen/ wheat flour mix was removed from LN and applied to flowering female cannabis plants. The pollination resulted in successful seed formation in all the flowers receiving treated pollen.”

Authors should show the data about the seed formation ability of treated pollen.

Miner comments

1, Line 79, “from Backues et al. [9]”

insert “(2010)”.

2, Line 187, “Baked wheat flower”

change flower to flour.

3, There are many mistakes in Ref. part.

e.g. html tags are remained in line, 252, 266, 290.

Line 300, “Nitorogen1” change to “Nitrogen”

Line 267, What does it mean “3218 LP -3231”?

Change to “3218-3231”.

Reviewer 4 Report

Gaudet et al present a short and well carried out manuscript. This manuscript is specifically valuable to scientists/industry interested in Cannabis crop preservation. There are a few minor issues that need to be taken care of before the manuscript can be accepted for publication. These are:

  1. Line 20, the authors did not follow up the study beyond 4 months, and yet they claim/suggest '....indefinite preservation of cannabis pollen'. This last part of their conclusion needs to be deleted in the Abstract and the main text!
  2. The authors use for hour(s), hours, hrs, in the text and legends to figures etc. They need to be consistent and use the normal 'h' throughout the text and figure legends.
  3. The References section also needs to be checked carefully to ensure the uniformity throughout.

Round 2

Reviewer 3 Report

All my concerns are addressed properly.

Author Response

Thank you very much for your valuable comments on our manuscript. We are glad that you found our manuscript suitable for publication in Plants.